# DETECTING OUT-OF-DISTRIBUTION INPUTS TO DEEP GENERATIVE MODELS USING TYPICALITY

## ABSTRACT

Recent work has shown that deep generative models can assign higher likelihood to out-of-distribution data sets than to their training data (Nalisnick et al., 2019; Choi et al., 2019). We posit that this phenomenon is caused by a mismatch between the model's typical set and its areas of high probability density. In-distribution inputs should reside in the former but not necessarily in the latter, as previous work has presumed (Bishop, 1994). To determine whether or not inputs reside in the typical set, we propose a statistically principled, easy-to-implement test using the empirical distribution of model likelihoods. The test is model agnostic and widely applicable, only requiring that the likelihood can be computed or closely approximated. We report experiments showing that our procedure can successfully detect the out-of-distribution sets in several of the challenging cases reported by Nalisnick et al. (2019).

## 1 INTRODUCTION

Recent work (Nalisnick et al., 2019; Choi et al., 2019; Shafaei et al., 2018) showed that a variety of deep generative models fail to distinguish training from out-of-distribution (OOD) data according to the model likelihood. This phenomenon occurs not only when the data sets are similar but also when they have dramatically different underlying semantics. For instance, *Glow* (Kingma & Dhariwal, 2018), a state-of-the-art normalizing flow, trained on CIFAR-10 will assign a higher likelihood to SVHN than to its CIFAR-10 training data (Nalisnick et al., 2019; Choi et al., 2019). This result is surprising since CIFAR-10 contains images of frogs, horses, ships, trucks, etc. and SVHN contains house numbers. A human would be very unlikely to confuse the two sets. These findings are also troubling from an algorithmic standpoint since higher OOD likelihoods break previously proposed methods for classifier validation (Bishop, 1994) and anomaly detection (Pimentel et al., 2014).

We conjecture that these high OOD likelihoods are evidence of the phenomenon of *typicality*.[1] Due to concentration of measure, a generative model will draw samples from its *typical set* (Cover & Thomas, 2012), a subset of the model's full support. However, the typical set may not necessarily intersect with regions of high probability *density*. For example, consider a $d$-dimensional isotropic Gaussian. Its highest density region is at its mode (the mean) but the typical set resides at a distance of $\sqrt{d}$ from the mode (Vershynin, 2018). Thus a point near the mode will have high likelihood while being extremely unlikely to be sampled from the model. We believe that deep generative models exhibit a similar phenomenon since, to return to the CIFAR-10 vs SVHN example, Nalisnick et al. (2019) showed that sampling from the model trained on CIFAR-10 never generates SVHN-looking images despite SVHN having higher likelihood.

Based on this insight, we propose that OOD detection should be done by checking if an input resides in the model's typical set, not just in a region of high density. Unfortunately it is impossible to analytically derive the regions of typicality for the vast majority of deep generative models. To define a widely applicable and scalable OOD-detection algorithm, we formulate Shannon (1948)'s entropy-based definition of typicality into a statistical hypothesis test. To ensure that the test is robust

---

[1]Choi et al. (2019) also consider typicality as an explanation but ultimately deem it not to be a crucial factor. Nalisnick et al. (2019) implicitly mention typicality in their discussion of the transformed representations' proximity to the mean and explicitly in a comment on Open Review: https://openreview.net/forum?id=H1xwNhCcYm&noteId=HkgLWfveT7

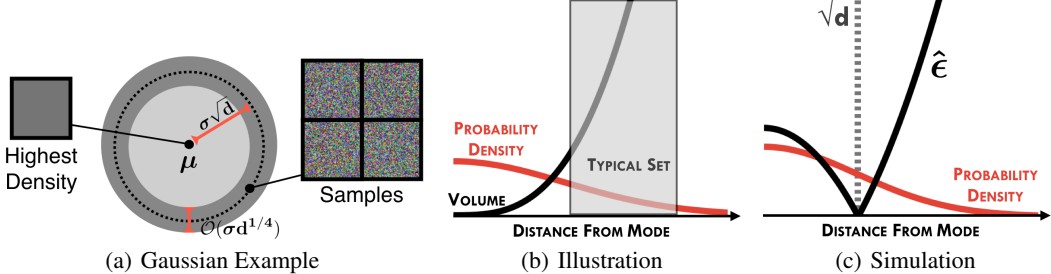

(a) Gaussian Example      (b) Illustration      (c) Simulation

Figure 1: *Typical Sets*. Subfigure (a) shows the example of a Gaussian with its mean located at the high-dimensional all-gray image. Subfigure (b) shows how the typical set arises due to the nature of high-dimensional integration. The figure is inspired by Betancourt (2017)'s similar illustration. Subfigure (c) shows our proposed method (Equation 3, higher $\hat{\epsilon}$ implies OOD) applied to a Gaussian simulation. The values have been re-scaled for purposes of visualization.

even in the low-data regime, we employ a bootstrap procedure (Efron & Tibshirani, 1994) to set the OOD-decision threshold. In the experiments, we demonstrate that our detection procedure succeeds in many of the challenging cases presented by Nalisnick et al. (2019). In addition to these successes, we also discuss failure modes that reveal drastic variability in OOD detection for the same data set pairs under different generative models. We highlight these cases to inspire future work.

## 2   BACKGROUND: TYPICAL SETS

The *typical set* of a probability distribution is the set whose elements have an information content sufficiently close to that of the expected information (Shannon, 1948). A formal definition follows.

**Definition 2.1** $(\epsilon, \mathbf{N})$-*Typical Set* (*Cover & Thomas, 2012*)   *For a distribution $p(\mathbf{x})$ with support $\mathbf{x} \in \mathcal{X}$, the $(\epsilon, N)$-typical set $\mathcal{A}_\epsilon^N[p(\mathbf{x})] \in \mathcal{X}^N$ is comprised of all $N$-length sequences that satisfy*

$$\mathbb{H}[p(\mathbf{x})] - \epsilon \leq \frac{1}{N} - \log p(\boldsymbol{x}_1, \ldots, \boldsymbol{x}_N) \leq \mathbb{H}[p(\mathbf{x})] + \epsilon$$

*where $\mathbb{H}[p(\mathbf{x})] = \int_{\mathcal{X}} p(\mathbf{x})[-\log p(\mathbf{x})]d\mathbf{x}$ and $\epsilon \in \mathbb{R}^+$ is a small constant.*

When the joint density in Definition 2.1 factorizes, we can write:

$$\mathbb{H}[p(\mathbf{x})] - \epsilon \ \leq \ \frac{1}{N} \sum_{n=1}^{N} - \log p(\boldsymbol{x}_n) \ \leq \ \mathbb{H}[p(\mathbf{x})] + \epsilon. \tag{1}$$

This is the definition we will use from here forward as we assume both training data and samples from our generative model are identically and independently distributed (i.i.d.). In this factorized form, the middle quantity can be interpreted as an $N$-sample empirical entropy: $1/N \sum_{n=1}^{N} - \log p(\boldsymbol{x}_n) = \hat{\mathbb{H}}^N[p(\mathbf{x})]$. The *asymptotic equipartition property* (AEP) (Cover & Thomas, 2012) states that this estimate will converge to the true entropy as $N \to \infty$.

To build intuition, let $p(\mathbf{x}) = \mathrm{N}(\mathbf{0}, \sigma^2 \mathbb{I})$ and consider its $(\epsilon, 1)$-typical set. Plugging in the relevant quantities to Equation 1 and simplifying, we have $\mathbf{x} \in \mathcal{A}_\epsilon^1[\mathrm{N}(\mathbf{0}, \sigma^2 \mathbb{I})]$ if $\frac{1}{2}|d - ||\mathbf{x} - \mu||_2^2/\sigma^2| \leq \epsilon$ where $d$ denotes dimensionality. See Appendix A.1 for a complete derivation. The inequality will hold for any choice of $\epsilon$ if $||\mathbf{x} - \mu||_2 = \sigma\sqrt{d}$. In turn, we can geometrically interpret $\mathcal{A}_\epsilon^1[\mathrm{N}(\mathbf{0}, \sigma^2 \mathbb{I})]$ as an annulus centered at $\mu$ with radius $\sigma\sqrt{d}$ and whose width is a function of $\epsilon$ (and $\sigma$). This is a well-known concentration of measure result often referred to as the *Gaussian Annulus Theorem* (Vershynin, 2018). Figure 1(a) illustrates a Gaussian centered on the all gray image (pixel value 128). We show that samples from this model never resemble the all gray image, despite it having the highest probability density, because they are drawn from the annulus. In Figure 1(b) we visualize the interplay between density and volume that gives rise to the typical set. The connection between typicality and concentration of measure can be stated formally as:

**Theorem 2.1** *Probability of the Typical Set* *(Cover & Thomas, 2012)* *For N sufficiently large, the typical set has probability*

$$P\left(\mathcal{A}_\epsilon^N[p(\mathbf{x})]\right) > 1 - \epsilon.$$

This result speaks to the central role of typical sets in compression: $\mathcal{A}_\epsilon^N$ is an efficient representation of $\mathcal{X}^N$ as it is sampled under $p(\mathbf{x})$.[2] Returning to the Gaussian example, we could 'compress' $\mathbb{R}^d$ under $N(\mathbf{0}, \sigma^2 \mathbb{I})$ to just the $\sigma\sqrt{d}$-radius annulus.[3]

## 3 A TYPICALITY TEST FOR OOD INPUTS

We next describe our core contribution: a reformulation of Definition 2.1 into a scalable goodness-of-fit test to determine if a batch of test data was likely drawn from a given deep generative model.

### 3.1 SETTING OF INTEREST: GOODNESS-OF-FIT TESTING FOR DEEP GENERATIVE MODELS

Assume we have a generative model $p(\mathbf{x}; \boldsymbol{\theta})$—with $\boldsymbol{\theta}$ denoting the parameters—that was trained on a data set $\boldsymbol{X} = \{\boldsymbol{x}_1, \ldots, \boldsymbol{x}_N\}$. Take $\mathbf{x}$ to be high-dimensional ($d > 500$) and $N$ to be sufficiently large ($N > 25,000$) so as to enable training a high-capacity neural-network parametrized model—a so-called 'deep generative model' (DGM). Furthermore, we assume that $p(\mathbf{x}; \theta)$ has a likelihood that can be evaluated either directly or closely approximated via Monte Carlo sampling. Examples of DGMs that meet these specifications include normalizing flows (Tabak & Turner, 2013) such as *Glow* (Kingma & Dhariwal, 2018), latent variable models such as *variational autoencoders* (VAEs) (Kingma & Welling, 2014; Rezende et al., 2014), and auto-regressive models such as *PixelCNN* (van den Oord et al., 2016). We do not consider implicit generative models (Mohamed & Lakshminarayanan, 2016) (such as GANs (Goodfellow et al., 2014)) due to their likelihood being difficult to even approximate.

The primary focus of this paper is in performing a *goodness-of-fit* (GoF) test (D'Agostino, 1986; Huber-Carol et al., 2012) for $p(\mathbf{x}; \boldsymbol{\theta})$. Specifically, given an $M$-sized *batch* of test observations $\widetilde{\boldsymbol{X}} = \{\tilde{\boldsymbol{x}}_1, \ldots, \tilde{\boldsymbol{x}}_M\}$ ($M \geq 1$), we desire to determine if $\widetilde{\boldsymbol{X}}$ was sampled (i.i.d.) from $p_{\boldsymbol{\theta}}$ or from some other distribution $q \neq p_{\boldsymbol{\theta}}$. We assume no knowledge of $q$, thus making our desired GoF test *omnibus* (Eubank & LaRiccia, 1992). The vast majority of GoF tests operate via the model's cumulative distribution function (CDF) and/or being able to compute an empirical distribution function (EDF) (Cramér, 1928; Massey Jr, 1951; Anderson & Darling, 1954; Stephens, 1974). However, the CDFs of DGMs are not available analytically, and numerical approximations are hopelessly slow due to the curse of dimensionality. Likewise, EDFs lose statistical strength exponentially as dimensionality grows (Wasserman, 2006). Our goal is to formulate a scalable test that does not rely on strong parametric assumptions (e.g. Chen & Xia (2019)) and has better computational properties than kernel-based alternatives (e.g. Liu et al. (2016)).

### 3.2 A HYPOTHESIS TEST FOR TYPICALITY

Returning to the results of Nalisnick et al. (2019) and Choi et al. (2019), the high-dimensionality of natural images ($d = 3072$ for CIFAR and SVHN) alone is enough to suspect the influence of phenomena akin to the Gaussian Annulus Theorem. Yet there are stronger parallels still: Nalisnick et al. (2019) showed that the all-black image has the highest density of any tested input to their FashionMNIST DGM, but this model is never observed to generate all-black images. Thus we are inspired to critique DGMs not via density but via *typical set membership*:

$$\text{if } \widetilde{\boldsymbol{X}} \in \mathcal{A}_\epsilon^M[p(\mathbf{x}; \boldsymbol{\theta})] \text{ then } \widetilde{\boldsymbol{X}} \sim p(\mathbf{x}; \boldsymbol{\theta}), \quad \text{otherwise } \widetilde{\boldsymbol{X}} \not\sim p(\mathbf{x}; \boldsymbol{\theta}). \tag{2}$$

The intuition is that if $\widetilde{\boldsymbol{X}}$ is indeed sampled from $p_{\boldsymbol{\theta}}$, then with high probability it must reside in the typical set (Theorem 2.1). To determine if $\widetilde{\boldsymbol{X}} \in \mathcal{A}_\epsilon^M[p(\mathbf{x}; \boldsymbol{\theta})]$, we can plug $\widetilde{\boldsymbol{X}}$ into Equation 1 as a

---

[2]While $\mathcal{A}_\epsilon^N$ is not the smallest high-probability set (Polonik, 1997) and therefore not the most efficient compression, its size is of the same order (Cover & Thomas, 2012).

[3]The reader may have noticed Theorem 2.1 requires 'N sufficiently large' but in the Gaussian example we assumed $N = 1$. For high-dimensional factorized likelihoods, $\log p(\mathbf{x}) = \sum_{j=1}^d \log p(x_j)$, and thus we can interpret Definition 2.1 as acting dimension-wise instead of across observations.

length $M$ sequence and check if the $\epsilon$-bound holds:

$$\texttt{if } \left| \frac{1}{M} \sum_{m=1}^{M} - \log p(\tilde{\boldsymbol{x}}_m; \boldsymbol{\theta}) - \mathbb{H}[p(\mathbf{x}; \boldsymbol{\theta})] \right| = \hat{\epsilon} \leq \epsilon \texttt{ then } \widetilde{\boldsymbol{X}} \in \mathcal{A}_\epsilon^M[p(\mathbf{x}; \boldsymbol{\theta})], \quad (3)$$

where $\hat{\epsilon}$ denotes the test statistic. We provide a sanity check for Equation 3 in Subfigure 1(c), showing $\hat{\epsilon}$ calculated for the high-dimensional Gaussian example described in Section 2. We see that $\hat{\epsilon}$ achieves its minimum value exactly at $\sqrt{d}$-distance from $\mathbf{128}$.

In Appendix A.2 we show that our test is consistent unless the alternative's typical set is a subset of $p_{\boldsymbol{\theta}}$'s: $\mathcal{A}_\epsilon^M[q(\mathbf{x})] \subseteq \mathcal{A}_\epsilon^M[p(\mathbf{x}; \boldsymbol{\theta})]$. This limitation is reasonable and expected given our fundamental assumption in Equation 2. Since the size of the typical set is upper bounded as a function of entropy—$\log |\mathcal{A}_\epsilon^M[p(\mathbf{x}; \boldsymbol{\theta})]| \leq M(\mathbb{H}[p(\mathbf{x}; \boldsymbol{\theta})] + \epsilon)$ (Cover & Thomas, 2012)—the model entropy determines the probability of type-II error: higher entropy implies a larger typical set, a larger set implies more chance of $\mathcal{A}_\epsilon^M[q] \subseteq \mathcal{A}_\epsilon^M[p_{\boldsymbol{\theta}}]$, and a higher degree of intersection leads to a better chance of incorrectly failing to reject $H_0 : \tilde{\boldsymbol{x}} \sim p_{\boldsymbol{\theta}}$. Yet it is not uncommon to sacrifice consistency for generality when testing GoF (e.g. Chi-square vs Kolmogorov-Smirnov tests (Haberman, 1988)).

## 3.3 IMPLEMENTATION DETAILS

In an ideal setting, we could mathematically derive the regions in $\mathcal{X}$ that correspond to the typical set (e.g. the Gaussian's annulus) and check if $\tilde{\boldsymbol{x}}$ resides within that region. Unfortunately, finding these regions is analytically intractable for neural-network-based generative models. A practical implementation of Equation 3 requires computing the entropy $\mathbb{H}[p(\mathbf{x}; \boldsymbol{\theta})]$ and the threshold $\epsilon$.

**Entropy Estimator** The entropy of DGMs is not available in closed-form and therefore we resort to the following sampling-based approximation. Recall from Subsection 2 that the AEP states that the sample entropy will converge to the true entropy as the number of samples grows. Since we have access to the model and can drawn a large number of samples from it, the empirical entropy should be a good approximation for the true model entropy:

$$\mathbb{H}[p(\mathbf{x}; \boldsymbol{\theta})] = \int_{\mathcal{X}} p(\mathbf{x}; \boldsymbol{\theta}) [- \log p(\mathbf{x}; \boldsymbol{\theta})] d\mathbf{x} \approx \frac{1}{S} \sum_{s=1}^{S} - \log p(\hat{\boldsymbol{x}}_s; \boldsymbol{\theta}) \quad (4)$$

where $\hat{\boldsymbol{x}}_s \sim p(\mathbf{x}; \boldsymbol{\theta})$. However, in preliminary experiments (reported in Appendix E.1) we observed markedly better OOD detection when using an alternative estimator known as the *resubstitution estimator* (Beirlant et al., 1997). This estimator uses the training set for calculating the expectation:

$$\mathbb{H}[p(\mathbf{x}; \boldsymbol{\theta})] \approx \hat{\mathbb{H}}_{\text{RESUB}}^N[p(\mathbf{x}; \boldsymbol{\theta})] = \frac{1}{N} \sum_{n=1}^{N} - \log p(\boldsymbol{x}_n; \boldsymbol{\theta}). \quad (5)$$

This approximation should be good as well since we assume $N$ to be large.[4]

**Setting the OOD-Threshold with the Bootstrap** Concerning the threshold $\epsilon$, we propose setting its value through simulation—by constructing a *bootstrap confidence interval* (BCI) (Efron, 1992; Arcones & Gine, 1992) for the null hypothesis $H_0 : \widetilde{\boldsymbol{X}} \in \mathcal{A}_\epsilon^M[p(\mathbf{x}; \boldsymbol{\theta})]$, with the alternative being $H_1 : \widetilde{\boldsymbol{X}} \notin \mathcal{A}_\epsilon^M[p(\mathbf{x}; \boldsymbol{\theta})]$. In a slight deviation from the tradition procedure for BCI construction, we assume the existence of a validation set $\boldsymbol{X}'$ that was held-out from $\boldsymbol{X}$ before training the generative model (just as is usually done for hyperparameter tuning). This is only to account for the generative model overfitting to the training set. From this validation set, we bootstrap sample $K$ 'new' data sets $\{\boldsymbol{X}'_k\}_{k=1}^K$ of size $M$ and then plug each into Equation 3 in place of $\widetilde{\boldsymbol{X}}$:

$$\left| \frac{1}{M} \sum_{m=1}^{M} - \log p(\boldsymbol{x}'_{k,m}; \boldsymbol{\theta}) - \hat{\mathbb{H}}_{\text{RESUB}}^N[p(\mathbf{x}; \boldsymbol{\theta})] \right| = \hat{\epsilon}_k \quad (6)$$

---

[4]The bias and variance of the resubstitution estimator are hard to characterize for DGMs. The work of Joe (1989) is most related, describing its properties under multivariate kernel density estimators.

where $\hat{\epsilon}_k$ is the estimate for the $k$th bootstrap sample. All $K$ estimates then form the bootstrap distribution $F(\epsilon) = \frac{1}{K} \sum_{k=1}^{K} \delta[\hat{\epsilon}_k]$. Calculating the $\alpha$-quantile of $F(\epsilon)$, which we denote as $\epsilon_\alpha^M$, determines the threshold at which we reject the null hypothesis with confidence-level $\alpha$ (Arcones & Gine, 1992). If we reject the null, then we decide that the sample does not reside in the typical set and therefore is OOD. The complete procedure is summarized in Algorithm 1 in Appendix B. Observe that nearly all of the computation can be performed offline before any test set is received, including all bootstrap simulations. The rejection threshold $\epsilon_\alpha^M$ depends on a particular $M$ and $\alpha$ setting, but these computations can be done in parallel across multiple machines. The most expensive test-time operation is obtaining $\log p(\tilde{\boldsymbol{x}}, \boldsymbol{\theta})$. After this is done, only an $\mathcal{O}(M)$ operation to sum the likelihoods is required.

## 4    RELATED WORK

**Goodness-of-Fit Tests**    As mentioned in Section 3.1, many of the traditional GoF tests are not applicable to the DGMs and high-dimensional data sets that we consider since CDFs and EDFs are both intractable in this setting. *Kernelized Stein discrepancy* (Chwialkowski et al., 2016; Liu et al., 2016) is a recently-proposed GoF test that can scale to the DGM regime, and we compare against it in the experiments. Several works have proposed GoF tests based on entropy (Gokhale, 1983; Parzen, 1990)—e.g. for normal (Vasicek, 1976), uniform (Dudewicz & Van Der Meulen, 1981), and exponential (Crzcgorzewski & Wirczorkowski, 1999) distributions. However, these tests are derived from maximum entropy results and not motivated from typicality. There are also directed GoF tests such as ones based on likelihood ratios (Neyman & Pearson, 1933; Wilks, 1938) or discrepancies such as KL divergence (Noughabi & Arghami, 2013). These tests require an explicit definition of $q$, which may be difficult in many DGM-appropriate scenarios. Yet the recent work of Ren et al. (2019) does apply likelihood ratios to PixelCNNs by constructing $q$ such that it models a background process (i.e. some perturbed version of the original data).

**Typical and Minimum Volume Sets**    We are aware of only two previous works that use a notion of typicality for GoF tests or OOD detection. Sabeti & Hst-Madsen (2019) propose a typicality framework based on minimum description length. They deem data as 'atypical' if it can be represented in less bits than one would expect under the generative model. While our frameworks share the same conceptual foundation, Sabeti & Hst-Madsen (2019)'s implementation relies on strong parametric assumptions and cannot be generalized to deep models (without drastic approximations). Choi et al. (2019), the second work, leverages normalizing flows to test for typicality by transforming the data to a normal distribution and then deeming points outside the annulus to be anomalous. This approach restricts the generative model to be a Gaussian normalizing flow whereas ours is applicable to any generative model with a computable likelihood. Our work is also related to the concept of *minimum volume* (MV) sets (Sager, 1979; Polonik, 1997; Garcia et al., 2003). MV sets have been used for GoF testing (Polonik, 1999; Glazer et al., 2012) and to detect outliers (Platt et al., 2001; Scott & Nowak, 2006; Clémençon et al., 2018). However, we are not aware of any work that scales MV-set-based methodologies to the degree required to be applicable to DGMs.

**Generative Models and Outlier Detection**    Probabilistic but non-test-based techniques have also been widely employed to discover outliers and anomalies (Pimentel et al., 2014). One of the most common is to use a (one-sided) threshold on the density function to classify points as OOD (Barnett et al., 1994); this idea is used in Tarassenko et al. (1995) Bishop (1994), and Parra et al. (1996), among others. Other work has applied more sophisticated techniques to density function evaluations—for instance, Clifton et al. (2014) applies extreme value theory. Yet this work and all others of which we are aware do not identify points with abnormally *high* density as OOD. Thus they would fail in the settings presented by Nalisnick et al. (2019). As for work focusing on DGMs in particular, most previous work proposes training improvements to make the model more robust. For instance, Hendrycks et al. (2019) show that robustness and uncertainty quantification w.r.t. outliers can be improved by exposing the model to an auxiliary data set (a proxy for OOD data) during training. As for post-training outlier and OOD detection, Choi et al. (2019) proposes using an ensemble of models to compute the *Watanabe-Akaike information criterion* (WAIC). However, there are no rigorous arguments for why WAIC should quantify GoF. Škvára et al. (2018) proposes using a VAE's conditional likelihood as an outlier criterion, finding that this works well only when the hyperparameters can be tuned using anomalous data. As far as we are aware, we are the first to

apply a hypothesis testing framework to the problem of OOD or anomaly detection for DGMs. As mentioned above, Ren et al. (2019) use likelihood ratios, but they do not perform a hypothesis test.

## 5 EXPERIMENTS

We now evaluate our typicality test's OOD detection abilities, focusing in particular on the image data set pairs highlighted by Nalisnick et al. (2019). We use the same three generative models as they did—Glow (Kingma & Dhariwal, 2018), PixelCNN (van den Oord et al., 2016), and Rosca et al. (2018)'s VAE architecture—attempting to replicate training and evaluation as closely as possible. See Appendix C for a full description of model architectures and training. See Appendix D for more details on evaluation. We consider the following baselines[5]; all statistical tests use $\alpha = 0.99$:

1. `t-test`: We apply a two-sample students' t-test to check for a difference in means in the empirical likelihoods. In terms of Equation 3, this baseline will reject for any $\epsilon > 0$, and thus we expect it to be overly conservative. Moreover, this test does not have access to validation data and therefore improvements upon it can be attributed to our bootstrap procedure.

2. `Kolmogorov-Smirnov test (KS-test)`: We apply a two-sample KS-test to the likelihood EDFs. This test is stronger than our typicality test since it is checking for equivalence in all moments whereas ours (and the t-test) is restricted to the first moment. In turn, this test has a greater computational complexity—$\mathcal{O}(M \log M)$ compared to $\mathcal{O}(M)$.

3. `Maximum Mean Discrepancy (MMD)`: We apply a two-sample MMD (Gretton et al., 2012) test to the data directly. Yet we incorporate the generative model by using a Fisher kernel (Jaakkola & Haussler, 1999). We also apply the same bootstrap procedure on validation data to construct the test statistic. MMD has greater runtime still at $\mathcal{O}(NMd)$. It also requires access to (a subset of) the training data at test-time, which is undesirable.

4. `Kernelized Stein Discrepancy (KSD)`: We apply KSD (Liu et al., 2016) to test for GoF to the generative model and again use a Fisher kernel and the bootstrap procedure on validation data. KSD has runtime $\mathcal{O}(M^2 d)$. While we have ignored the construction of the kernel in the runtime analysis, KSD is the most costly since it requires computing three model gradients.

5. `Annulus Method`: We use a modified version of Choi et al. (2019)'s annulus method applied to Gaussian normalizing flows. Like them, we classify something as OOD based on its distance to the sphere with radius $\sqrt{d}$. This is essentially performing our test but via closed-form expressions for entropy made available by the Gaussian base distribution. We use the same bootstrap procedure on validation data to set the 'slack' variable $\epsilon$.

**Grayscale Images** We first evaluate our typicality test on grayscale images. We trained a Glow, PixelCNN, and VAE each on the FashionMNIST training split and tested OOD detection using the FashionMNIST, MNIST, and NotMNIST test splits. We use the FashionMNIST test split to evaluate for type-I error (incorrect rejection of the null) and the MNIST and NotMNIST splits for type-II error (incorrect rejection of the alternative). In Figure 2 we show the empirical distribution of likelihoods over each data set for each model. We see the same phenomenon as reported by Nalisnick et al. (2019)—namely, that the MNIST OOD test set (green) has a higher likelihood than the training set (black). Lower-sided thresholding (Bishop, 1994) would clearly fail to detect the OOD sets. Table 1 reports a comparison against baselines, showing the fraction of $M$-sized batches classified as OOD. The IN-DIST. column reports the value for the FashionMNIST test set and ideally this number should be 0.00; any deviation from zero corresponds to type-I error. Conversely, the MNIST and NOTMNIST columns should be 1.00, and any deviation corresponds to type-II error. We see that for $M = 2$ all tests find it hard to reject the null hypothesis, which is not surprising given the overlap in the histograms in Figure 2. The exceptions are the annulus method for NotMNIST-Glow (96%), the typicality test for MNIST-PixelCNN (56%), and all methods except KS-test for NotMNIST-VAE. One failure mode for almost all methods is NotMNIST for the PixelCNN. None of the likelihood-based tests can distinguish NotMNIST as OOD due to the near perfect overlap in histograms shown in Figure 2(b). KSD and especially MMD are able to perform better in this case due to having access to the original feature-space representations (in addition to the generative model). Yet, surprisingly, KSD and MMD perform comparatively poorly for MNIST, especially at $M = 10$ and $M = 25$. The

---

[5]We could not replicate the performance of WAIC as reported by Choi et al. (2019). See Appendix E.2.

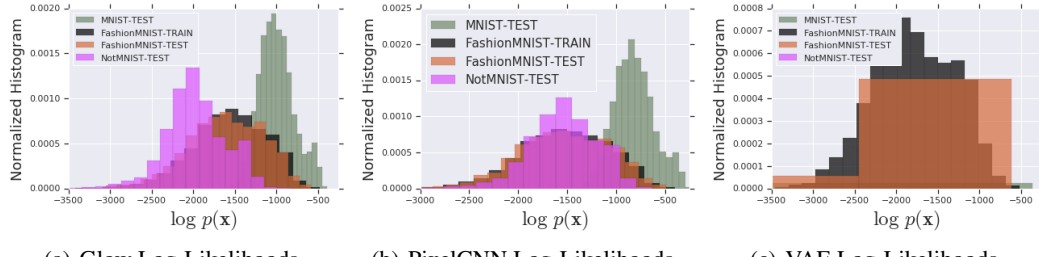

(a) Glow Log-Likelihoods     (b) PixelCNN Log-Likelihoods     (c) VAE Log-Likelihoods

Figure 2: *Empirical Distribution of Likelihoods*. The above figure shows the histogram of log-likelihoods for FashionMNIST (train, test), MNIST (test), and NotMNIST (test) for the (a) Glow, (b) PixelCNN, and (c) VAE.

Table 1: *Grayscale Images: Fraction of $M$-Sized Batches Classified as OOD*. The in-distribution column reflects type-I error and the MNIST and NotMNIST columns reflect type-II.

| | | M = 2 | | | M = 10 | | | M = 25 | |
|---|---|---|---|---|---|---|---|---|---|
| METHOD | IN-DIST. | MNIST | NOTMNIST | IN-DIST. | MNIST | NOTMNIST | IN-DIST. | MNIST | NOTMNIST |
| | | | *Glow Trained on FashionMNIST* | | | | | | |
| **Typicality Test** | $0.02_{\pm.01}$ | $0.14_{\pm.10}$ | $0.08_{\pm.04}$ | $0.02_{\pm.02}$ | $\mathbf{1.00}_{\pm.00}$ | $0.69_{\pm.11}$ | $0.01_{\pm.00}$ | $\mathbf{1.00}_{\pm.00}$ | $\mathbf{1.00}_{\pm.00}$ |
| $t$-Test | $0.01_{\pm.00}$ | $0.08_{\pm.00}$ | $0.06_{\pm.00}$ | $\mathbf{0.01}_{\pm.00}$ | $\mathbf{1.00}_{\pm.00}$ | $0.67_{\pm.01}$ | $0.01_{\pm.00}$ | $\mathbf{1.00}_{\pm.00}$ | $0.99_{\pm.00}$ |
| KS-Test | $\mathbf{0.00}_{\pm.00}$ | $0.00_{\pm.00}$ | $0.00_{\pm.00}$ | $\mathbf{0.01}_{\pm.00}$ | $\mathbf{1.00}_{\pm.00}$ | $0.61_{\pm.01}$ | $\mathbf{0.00}_{\pm.00}$ | $\mathbf{1.00}_{\pm.00}$ | $0.98_{\pm.01}$ |
| Max Mean Dis. | $0.05_{\pm.02}$ | $\mathbf{0.17}_{\pm.06}$ | $0.04_{\pm.03}$ | $0.02_{\pm.02}$ | $0.63_{\pm.12}$ | $0.37_{\pm.24}$ | $0.04_{\pm.04}$ | $\mathbf{1.00}_{\pm.00}$ | $\mathbf{1.00}_{\pm.00}$ |
| Kern. Stein Dis. | $0.05_{\pm.05}$ | $0.16_{\pm.14}$ | $0.01_{\pm.01}$ | $\mathbf{0.01}_{\pm.01}$ | $0.21_{\pm.11}$ | $0.01_{\pm.00}$ | $0.02_{\pm.03}$ | $0.76_{\pm.21}$ | $0.00_{\pm.00}$ |
| Annulus Method | $0.01_{\pm.01}$ | $0.00_{\pm.00}$ | $\mathbf{0.96}_{\pm.03}$ | $0.02_{\pm.00}$ | $0.00_{\pm.00}$ | $\mathbf{1.00}_{\pm.00}$ | $0.03_{\pm.03}$ | $0.00_{\pm.00}$ | $\mathbf{1.00}_{\pm.00}$ |
| | | | *PixelCNN Trained on FashionMNIST* | | | | | | |
| **Typicality Test** | $0.03_{\pm.01}$ | $\mathbf{0.56}_{\pm.13}$ | $0.01_{\pm.00}$ | $0.04_{\pm.02}$ | $\mathbf{1.00}_{\pm.00}$ | $0.01_{\pm.01}$ | $0.05_{\pm.03}$ | $\mathbf{1.00}_{\pm.00}$ | $0.01_{\pm.01}$ |
| $t$-Test | $0.01_{\pm.00}$ | $0.23_{\pm.00}$ | $0.00_{\pm.00}$ | $\mathbf{0.01}_{\pm.00}$ | $\mathbf{1.00}_{\pm.00}$ | $0.00_{\pm.00}$ | $\mathbf{0.02}_{\pm.00}$ | $\mathbf{1.00}_{\pm.00}$ | $0.00_{\pm.00}$ |
| KS-Test | $\mathbf{0.00}_{\pm.00}$ | $0.00_{\pm.00}$ | $0.00_{\pm.00}$ | $0.02_{\pm.00}$ | $\mathbf{1.00}_{\pm.00}$ | $0.00_{\pm.00}$ | $0.04_{\pm.00}$ | $\mathbf{1.00}_{\pm.00}$ | $0.01_{\pm.00}$ |
| Max Mean Dis. | $0.02_{\pm.00}$ | $0.05_{\pm.01}$ | $\mathbf{0.36}_{\pm.05}$ | $0.05_{\pm.02}$ | $0.27_{\pm.06}$ | $\mathbf{1.00}_{\pm.00}$ | $0.06_{\pm.04}$ | $0.59_{\pm.10}$ | $\mathbf{1.00}_{\pm.00}$ |
| Kern. Stein Dis. | $0.01_{\pm.00}$ | $0.05_{\pm.02}$ | $0.08_{\pm.03}$ | $0.02_{\pm.01}$ | $0.29_{\pm.14}$ | $0.61_{\pm.20}$ | $0.05_{\pm.02}$ | $0.70_{\pm.11}$ | $0.99_{\pm.01}$ |
| | | | *VAE Trained on FashionMNIST* | | | | | | |
| **Typicality Test** | $0.03_{\pm.01}$ | $\mathbf{0.37}_{\pm.05}$ | $0.99_{\pm.00}$ | $0.04_{\pm.02}$ | $0.94_{\pm.02}$ | $\mathbf{1.00}_{\pm.00}$ | $0.04_{\pm.03}$ | $0.96_{\pm.01}$ | $\mathbf{1.00}_{\pm.00}$ |
| $t$-Test | $0.01_{\pm.00}$ | $0.20_{\pm.00}$ | $0.99_{\pm.00}$ | $\mathbf{0.02}_{\pm.00}$ | $0.93_{\pm.00}$ | $\mathbf{1.00}_{\pm.00}$ | $0.02_{\pm.00}$ | $0.96_{\pm.00}$ | $\mathbf{1.00}_{\pm.00}$ |
| KS-Test | $\mathbf{0.00}_{\pm.00}$ | $0.00_{\pm.00}$ | $0.00_{\pm.00}$ | $0.02_{\pm.00}$ | $\mathbf{1.00}_{\pm.00}$ | $\mathbf{1.00}_{\pm.00}$ | $0.02_{\pm.00}$ | $\mathbf{1.00}_{\pm.00}$ | $\mathbf{1.00}_{\pm.00}$ |
| Max Mean Dis. | $0.03_{\pm.02}$ | $0.16_{\pm.07}$ | $0.73_{\pm.01}$ | $0.03_{\pm.04}$ | $0.41_{\pm.16}$ | $\mathbf{1.00}_{\pm.00}$ | $\mathbf{0.01}_{\pm.01}$ | $0.64_{\pm.05}$ | $\mathbf{1.00}_{\pm.00}$ |
| Kern. Stein Dis. | $0.04_{\pm.01}$ | $0.05_{\pm.01}$ | $0.74_{\pm.00}$ | $0.11_{\pm.04}$ | $0.17_{\pm.01}$ | $\mathbf{1.00}_{\pm.00}$ | $0.06_{\pm.04}$ | $0.37_{\pm.03}$ | $\mathbf{1.00}_{\pm.00}$ |

annulus method was unable to detect MNIST, which we found surprising given its close relationship to our typicality test, which does perform well. Yet Choi et al. (2019) note that Gaussian normalizing flows do not necessarily make the latent space normally distributed, and our typicality test may be able to use information from the volume element that is not available to the annulus method.

**Natural Images** We next turn to data sets of natural images—in particular SVHN, CIFAR-10, and ImageNet. We train Glow on SVHN, CIFAR-10, and ImageNet and use the two non-training sets for OOD evaluation. We found using MMD and KSD to be too expensive to make OOD decisions in an online system. Table 2 reports the fraction of $M$-sized batches classified as OOD. We see that our method (first row, bolded) is able to easily detect the OOD sets for SVHN, rejecting size-two batches at the rate of 98%+ while having only 1% type-I error. Performance on the CIFAR-10-trained model is good as well with 42%+ of OOD batches detected at $M = 2$ and 100% at $M = 10$ (type-I error at 1% in both cases). The hardest case is Glow trained on ImageNet: the KS-test performed best at $M = 25$ with 89%, followed by the t- and typicality tests at 72% and 74% respectively. The annulus method again had varying performance, being conspicuously inferior at detecting SVHN for the CIFAR and ImageNet models while having the best performance on ImageNet for the CIFAR model. We report additional results in Appendix E.3 for our method, showing performance for all $M \in [1, 150]$ and when using CIFAR-100 as an OOD set.

Lastly, we report two challenging cases worthy of note and further attention. Figure 3(a) shows our method applied to Glow when trained on CIFAR-10, tested on CIFAR-100. The $y$-axis again shows fraction of batches reported as OOD and the $x$-axis the batch size $M$. Even at $M = 150$ our method classifies only $\sim 20\%$ of batches as OOD. Yet this result is not surprising given that CIFAR-

Table 2: *Natural Images: Fraction of $M$-Sized Batches Classified as OOD.*

| METHOD | M = 2 | | | M = 10 | | | M = 25 | | |
|---|---|---|---|---|---|---|---|---|---|
| | SVHN | CIFAR-10 | IMAGENET | SVHN | CIFAR-10 | IMAGENET | SVHN | CIFAR-10 | IMAGENET |
| *Glow Trained on SVHN* | | | | | | | | | |
| **Typicality Test** | 0.01±.00 | **0.98**±.00 | **1.00**±.00 | **0.00**±.00 | **1.00**±.00 | **1.00**±.00 | 0.02±.00 | **1.00**±.00 | **1.00**±.00 |
| *t*-Test | **0.00**±.00 | 0.95±.00 | **1.00**±.00 | 0.04±.00 | **1.00**±.00 | **1.00**±.00 | 0.03±.00 | **1.00**±.00 | **1.00**±.00 |
| KS-Test | **0.00**±.00 | 0.00±.00 | 0.00±.00 | 0.08±.00 | **1.00**±.00 | **1.00**±.00 | 0.03±.00 | **1.00**±.00 | **1.00**±.00 |
| Annulus Method | 0.02±.01 | 0.70±.05 | **1.00**±.00 | 0.02±.01 | **1.00**±.00 | **1.00**±.00 | **0.00**±.00 | **1.00**±.00 | **1.00**±.00 |
| *Glow Trained on CIFAR-10* | | | | | | | | | |
| **Typicality Test** | 0.42±.09 | 0.01±.01 | 0.64±.04 | **1.00**±.00 | 0.01±.01 | **1.00**±.00 | **1.00**±.00 | 0.01±.01 | **1.00**±.00 |
| *t*-Test | 0.44±.01 | 0.01±.00 | 0.65±.00 | **1.00**±.00 | 0.02±.00 | **1.00**±.00 | **1.00**±.00 | 0.02±.00 | **1.00**±.00 |
| KS-Test | 0.00±.00 | **0.00**±.00 | 0.00±.00 | **1.00**±.00 | 0.01±.00 | 0.98±.00 | **1.00**±.00 | 0.01±.00 | **1.00**±.00 |
| Annulus Method | 0.09±.03 | 0.02±.00 | **0.87**±.05 | 0.19±.01 | 0.03±.00 | **1.00**±.00 | 0.35±.02 | 0.04±.00 | **1.00**±.00 |
| *Glow Trained on ImageNet* | | | | | | | | | |
| **Typicality Test** | **0.78**±.08 | 0.02±.01 | 0.01±.00 | **1.00**±.00 | 0.20±.06 | 0.01±.01 | **1.00**±.00 | 0.74±.05 | 0.01±.01 |
| *t*-Test | 0.76±.00 | 0.02±.00 | 0.01±.00 | **1.00**±.00 | 0.18±.01 | 0.01±.00 | **1.00**±.00 | 0.72±.01 | 0.01±.00 |
| KS-Test | 0.00±.00 | 0.00±.00 | 0.00±.00 | **1.00**±.00 | **0.29**±.01 | 0.01±.00 | **1.00**±.00 | **0.89**±.01 | 0.02±.00 |
| Annulus Method | 0.00±.00 | **0.03**±.00 | 0.02±.01 | 0.02±.02 | 0.15±.04 | 0.02±.00 | 0.16±.04 | 0.57±.12 | 0.02±.00 |

10 is a subset of CIFAR-100, which means that our test's subset assumptions for consistency are violated. More interesting is the case of Glow trained on CelebA, tested on CIFAR-10 and CIFAR-100. Figure 3(b) shows the histogram of log-likelihoods: all distributions peak at nearly the same value. The distribution of $\epsilon$ observed during the bootstrap procedure ($M = 200$) is shown in Figure 3(c), with the red and black dotted lines denoting $\hat{\epsilon}$ computed using the whole set. We see that $\hat{\epsilon}$ for the OOD set is even less than the in-distribution's, meaning that it would be impossible to reliably reject the OOD data while not rejecting the in-distribution test set as well. Interestingly, PixelCNN and VAE do not have as dramatic of an overlap in likelihoods—a phenomenon that can also be observed in Figure 2—which implies that the ability to detect OOD sets does not only depend on the data involved but the models as well. Some models may have likelihood functions that are reliably discriminative, and this presents an intriguing area for future work.

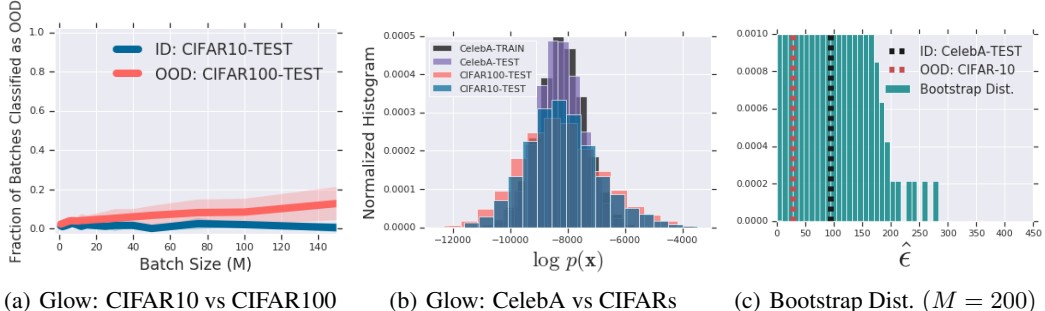

(a) Glow: CIFAR10 vs CIFAR100    (b) Glow: CelebA vs CIFARs    (c) Bootstrap Dist. ($M = 200$)

Figure 3: *Challenging Cases: CIFAR-10 vs CIFAR-100, CelebA vs CIFAR's.*

## 6   DISCUSSION AND CONCLUSIONS

We have presented a model-agnostic and computationally efficient statistical test for OOD inputs derived from the concept of typical sets. In the experiments we showed that the proposed test is especially well-suited to DGMs, identifying the OOD set for SVHN vs CIFAR-10 vs ImageNet (Nalisnick et al., 2019) with high accuracy (while maintaining $\leq 1\%$ type-I error). In this work we used the null hypothesis $H_0 : \widetilde{X} \in \mathcal{A}_\epsilon^M$, which was necessary since we assumed access to only one training data set. One avenue for future work is to use auxiliary data sets (Hendrycks et al., 2019) to construct a test statistic for the null $H_0 : \widetilde{X} \notin \mathcal{A}_\epsilon^M$, as would be proper for safety-critical applications. In our experiments we also noticed two cases—PixelCNN trained on FashionMNIST, tested on NotMNIST and Glow trained on CelebA, tested on CIFAR—in which the empirical distributions of in- and out-of-distribution likelihoods matched near perfectly. Thus use of the likelihood distribution produced by DGMs has a fundamental limitation that is seemingly worse than what was reported by Nalisnick et al. (2019).

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

# A    THEORETICAL PROPERTIES

## A.1    CONNECTION BETWEEN ENTROPY AND GAUSSIAN ANNULUS

For the sake of completeness, we make explicit the connection between Definition 2.1 and the Gaussian annulus example. Plugging in the spherical Gaussian's entropy and density function into Equation 1, we have:

$$
\begin{aligned}
\epsilon &\geq \left| d\log\sigma + \frac{d}{2}(1+\log 2\pi) - d\log\sigma - \frac{d}{2}\log 2\pi - \frac{1}{N}\sum_n \frac{||\boldsymbol{x}_n - \mu||_2^2}{2\sigma^2}\right| \\
&= \frac{1}{2}\left| d - \frac{1}{N}\sum_n \frac{||\boldsymbol{x}_n - \mu||_2^2}{\sigma^2}\right|.
\end{aligned}
\tag{7}
$$

For $N = 1$, we see that any point $\mathbf{x}$ that satisfies $||\mathbf{x} - \mu||_2 = \sigma\sqrt{d}$ guarantees the bound for any $\epsilon$:

$$
\epsilon \geq \frac{1}{2}\left| d - \frac{(\sigma\sqrt{d})^2}{\sigma^2}\right| = \frac{1}{2}\left| d - \frac{\sigma^2 d}{\sigma^2}\right| = 0.
\tag{8}
$$

Recalling Figure 1(a), $\sigma\sqrt{d}$ is exactly the radius of the annulus at which the Gaussian's mass concentrates. Of course as $\epsilon$ grows, points further from or nearer to the mean than $\sigma\sqrt{d}$ are included as typical. The behavior for finite $N$ is harder to characterize, as the definition is essential testing the $\epsilon$-bound for the average squared norm. Yet we know that for large samples $N \to \infty$,

$$
\frac{1}{N}\sum_n \frac{||\boldsymbol{x}_n - \mu||_2^2}{\sigma^2} \to \frac{\mathbb{E}[||\mathbf{x} - \mu||_2^2]}{\sigma^2} = d,
$$

which again allows the bound to hold for any $\epsilon$.

## A.2    CONSISTENCY OF THE TEST

Below we show that the test presented in Section 3.2 is consistent unless $\mathcal{A}_\epsilon^M[q(\mathbf{x})] \subseteq \mathcal{A}_\epsilon^M[p(\mathbf{x};\boldsymbol{\theta})]$.

**Proposition A.1** $\mathbf{p_\theta} \overset{\mathbf{d}}{=} \mathbf{q}$ *When* $\tilde{\mathbf{X}} \sim p(\mathbf{x};\boldsymbol{\theta})$, *the test statistic*

$$
\left| \frac{1}{M}\sum_{m=1}^M -\log p(\tilde{\boldsymbol{x}}_m;\boldsymbol{\theta}) - \mathbb{H}[p(\mathbf{x};\boldsymbol{\theta})] \right| = \hat{\epsilon} \xrightarrow{p} 0 \ \text{as } M \to \infty.
$$

*Proof*: The result follows directly from the AEP (Cover & Thomas, 2012). Alternatively, as $M \to \infty$, $\frac{1}{M}\sum_{m=1}^M -\log p(\tilde{\boldsymbol{x}}_m;\boldsymbol{\theta}) \to -\mathbb{E}[\log p(\tilde{\mathbf{x}};\boldsymbol{\theta})]$. We then have

$$
|-\mathbb{E}[\log p(\tilde{\mathbf{x}};\boldsymbol{\theta})] - \mathbb{H}[p(\mathbf{x};\boldsymbol{\theta})]| = \mathrm{KLD}\left[p(\mathbf{x};\boldsymbol{\theta})||p(\mathbf{x};\boldsymbol{\theta})\right] = 0.
$$

**Proposition A.2** $\mathbf{p_\theta} \neq \mathbf{q}$ *When* $\tilde{\mathbf{X}} \sim q(\mathbf{x})$ *such that* $p(\mathbf{x};\boldsymbol{\theta}) \neq q(\mathbf{x})$ *and* $\mathcal{A}_\epsilon^M[q(\mathbf{x})] \not\subseteq \mathcal{A}_\epsilon^M[p(\mathbf{x};\boldsymbol{\theta})]$, *the test statistic*

$$
\left| \frac{1}{M}\sum_{m=1}^M -\log p(\tilde{\boldsymbol{x}}_m;\boldsymbol{\theta}) - \mathbb{H}[p(\mathbf{x};\boldsymbol{\theta})] \right| > 0 \ \text{as } M \to \infty.
$$

*Proof (By Contradiction)*: As $M \to \infty$, $\frac{1}{M}\sum_{m=1}^M -\log p(\tilde{\boldsymbol{x}}_m;\boldsymbol{\theta}) \to -\mathbb{E}_q[\log p(\tilde{\mathbf{x}};\boldsymbol{\theta})]$. Assume that $|-\mathbb{E}_q[\log p(\tilde{\mathbf{x}};\boldsymbol{\theta})] - \mathbb{H}[p(\mathbf{x};\boldsymbol{\theta})]| = 0$ and that $\mathcal{A}_\epsilon^M[q(\mathbf{x})] \not\subseteq \mathcal{A}_\epsilon^M[p(\mathbf{x};\boldsymbol{\theta})]$. Then from Definition 2.1 we have

$$
\mathbb{H}[p(\mathbf{x})] - \epsilon \ \leq \ -\mathbb{E}_q[\log p(\tilde{\mathbf{x}};\boldsymbol{\theta})] \ \leq \ \mathbb{H}[p(\mathbf{x})] + \epsilon,
$$

which implies that $\mathcal{A}_\epsilon^M[q(\mathbf{x})] \subseteq \mathcal{A}_\epsilon^M[p(\mathbf{x};\boldsymbol{\theta})]$ for sufficiently large $M$. This contradicts our assumption that $\mathcal{A}_\epsilon^M[q(\mathbf{x})] \not\subseteq \mathcal{A}_\epsilon^M[p(\mathbf{x};\boldsymbol{\theta})]$ and therefore $|-\mathbb{E}_q[\log p(\tilde{\mathbf{x}};\boldsymbol{\theta})] - \mathbb{H}[p(\mathbf{x};\boldsymbol{\theta})]| > 0$.

---

**Algorithm 1** A Bootstrap Test for Typicality

---

**Input**: Training data $\boldsymbol{X}$, validation data $\boldsymbol{X'}$, trained model $p(\mathbf{x}; \boldsymbol{\theta})$, number of bootstrap samples $K$, significance level $\alpha$, $M$-sized batch of possibly OOD inputs $\widetilde{\boldsymbol{X}}$.

*Offline prior to deployment*
**1. Compute** $\hat{\mathbb{H}}^N[p(\mathbf{x}; \boldsymbol{\theta})] = \frac{-1}{N} \sum_{n=1}^{N} \log p(\boldsymbol{x}_n; \boldsymbol{\theta})$.
**2. Sample** $K$ $M$-sized data sets from $\mathbf{X'}$ using bootstrap resampling.
**3. For all** $k \in [1, K]$:
    **Compute** $\hat{\epsilon}_k = \left| \frac{-1}{M} \sum_{m=1}^{M} \log p(\boldsymbol{x}'_{k,m}; \boldsymbol{\theta}) - \hat{\mathbb{H}}^N[p(\mathbf{x}; \boldsymbol{\theta})] \right|$    *(Equation 6)*
**4. Set** $\epsilon_\alpha^M = \texttt{quantile}(F(\epsilon), \alpha)$    *(e.g. $\alpha = .99$)*

*Online during deployment*
**If** $\left| \frac{-1}{M} \sum_{m=1}^{M} \log p(\tilde{\boldsymbol{x}}_m) - \hat{\mathbb{H}}^N[p(\mathbf{x}; \boldsymbol{\theta})] \right| > \epsilon_\alpha^M$:
    **Return** $\widetilde{\mathbf{X}}$ `is out-of-distribution`
**Else**:
    **Return** $\widetilde{\mathbf{X}}$ `is in-distribution`

---

# B    ALGORITHMIC IMPLEMENTATION

The pseudocode of the procedure is described in Algorithm 1.

# C    GENERATIVE MODEL DETAILS

**Glow**    Our *Glow* (Kingma & Dhariwal, 2018) implementation was derived from OpenAI's open source repository[6] and modified following the specifications in Appendix A of Nalisnick et al. (2019). All versions were trained with RMSProp, batch size of 32, with a learning rate of $1 \times 10^{-5}$ for 100k steps and decayed by a factor of 2 after 80k and 90k steps. All priors were chosen to be standard Normal distributions. We follow Nalisnick et al. (2019)'s zero-initialization strategy (last coupling layer set to zero) and in turn did not apply any normalization. Similarly, our convolutional layers were initialized by sampling from the same truncated Normal distribution (Nalisnick et al., 2019). For our FashionMNIST experiment, Glow had two blocks of 16 affine coupling layers (ACLs) (Dinh et al., 2017). The spatial dimension was only squeezed between blocks. For the SVHN, CIFAR-10, and ImageNet models, we used three blocks of 8 ACLs with multi-scale factorization occurring between each block. All ACL transformations used a three-layer highway network. 200 hidden units were used for fashionMNIST and 400 for all other data sets.

**PixelCNN**    We trained a GatedPixelCNN (van den Oord et al., 2016) using Adam ($1 \times 10^{-4}$ initial learning rate, decayed by $1/3$ at steps 80k and 90k, 100k total steps) for FashionMNIST and RMSProp ($1 \times 10^{-4}$ initial learning rate, decayed by $1/3$ at steps 120k, 180k, and 195k, 200k total steps) for all other data sets. The FashionMNIST network had 5 gated layers (32 features) and a 256-sized skip connection. All other networks used 15 gated layers (128 features) and a 1024-sized skip connection

**Variational Autoencoder**    We used the convolutional decoder VAE (Kingma & Welling, 2014) variant described by Rosca et al. (2018). For Fashion MNIST, the decoder contained three convolutional layers with filter sizes 32, 32, and 256 and stides of 2, 2, and 1. Training was done again via RMSProp ($1 \times 10^{-4}$ initial learning rate, no decay, 200k total steps). For all other models, we followed the specifications in Rosca et al. (2018) Appendix K.

---

[6]https://github.com/openai/glow

## D    EXPERIMENTAL DETAILS

**MMD and KSD Kernels**    We found that MMD and KSD only had good performance when using the Fisher kernel (Jaakkola & Haussler, 1999): $k(\boldsymbol{x}_i, \boldsymbol{x}_j) = (\nabla_{\boldsymbol{\theta}} \log p(\boldsymbol{x}_i; \boldsymbol{\theta}))^T \nabla_{\boldsymbol{\theta}} \log p(\boldsymbol{x}_j; \boldsymbol{\theta})$. All other kernels attempted required substantial tuning to the scale parameters and we did not want to assume access to enough data to perform this tuning. The ineffectiveness of MMD on pixel-space has been noted previously (Bikowski et al., 2018). Furthermore, we found the memory cost of implementing the traditional Fisher kernel to be quite costly for Glow, each vector having 2million+ elements. Hence in the experiments we use the kernel modified such that the derivative is taken w.r.t. the input (making it the likelihood score): $k'(\boldsymbol{x}_i, \boldsymbol{x}_j) = (\nabla_{\boldsymbol{x}_i} \log p(\boldsymbol{x}_i; \boldsymbol{\theta}))^T \nabla_{\boldsymbol{x}_j} \log p(\boldsymbol{x}_j; \boldsymbol{\theta})$.

**Data Set Splits and Bootstrap Re-Samples**    For each data set we used the canonical train-test splits. To construct the validation set and perform bootstrapping, we extracted $5,000$ samples from the test split and bootstrap sampled (with replacement) $K = 50$ data sets to calculate $F(\epsilon)$. We didn't find using $K > 50$ to markedly change performance. We then extracted another $5,000$ samples from the test split, divided them into $M$-sized batches, and classified each other as OOD or not according to the various tests. We repeated this whole process 10 times, randomizing the instances in the validation and testing splits, in order to compute the means and standard deviations that are reported in Tables 1 and 2.

$\alpha$**-Level**    In preliminary experiments, we did not find a notable difference in type-II error when using $\alpha = 0.95$ vs $\alpha = 0.99$. Using the latter slightly improved type-I error and thus we used that value for all experiments and all methods.

## E    ADDITIONAL RESULTS

### E.1    COMPARING ENTROPY ESTIMATORS

In the tables below, we report results comparing the two entropy estimators considered—the Monte Carlo approximation with samples from the model (Equation 4) vs the resubstitution estimator (Equation 5). We see that the samples-based estimator performs better in only one setting, FashionMNIST vs MNIST at $M = 2$. In all other cases, the resubstitution estimator performs equally well or better. In fact, the samples-based estimator could not detect NotMNIST as OOD at all, having $0\%$ even at $M = 10$ and $M = 25$. This inferior performance is mostly due to the distribution of likelihoods being more diffuse when computed with samples. We suspect improvements to the generative models that enable them to better capture the true generative process will in turn improve the MC sample-based estimator.

Table 3: *Grayscale Images: Fraction of $M$-Sized Batches Classified as OOD*. The in-distribution column reflects type-I error and the MNIST and NotMNIST columns reflect type-II.

| METHOD | M = 2 | | | M = 10 | | | M = 25 | | |
|---|---|---|---|---|---|---|---|---|---|
| | IN-DIST. | MNIST | NOTMNIST | IN-DIST. | MNIST | NOTMNIST | IN-DIST. | MNIST | NOTMNIST |
| *Glow Trained on FashionMNIST* | | | | | | | | | |
| **Typicality Test w/ Data** | **0.02**±.01 | 0.14±.10 | **0.08**±.04 | **0.02**±.02 | **1.00**±.00 | **0.69**±.11 | **0.01**±.00 | **1.00**±.00 | **1.00**±.00 |
| Typicality Test w/ Samples | **0.02**±.01 | **0.44**±.17 | 0.00±.00 | 0.03±.03 | **1.00**±.00 | 0.00±.00 | 0.06±.05 | **1.00**±.00 | 0.00±.00 |

Table 4: *Natural Images: Fraction of $M$-Sized Batches Classified as OOD*.

| METHOD | M = 2 | | | M = 10 | | | M = 25 | | |
|---|---|---|---|---|---|---|---|---|---|
| | SVHN | CIFAR-10 | IN-DIST. | SVHN | CIFAR-10 | IN-DIST. | SVHN | CIFAR-10 | IN-DIST. |
| *Glow Trained on ImageNet* | | | | | | | | | |
| **Typicality Test w/ Data** | **0.78**±.08 | **0.02**±.01 | **0.01**±.00 | **1.00**±.00 | **0.20**±.06 | **0.01**±.01 | **1.00**±.00 | **0.74**±.05 | **0.01**±.01 |
| Typicality Test w/ Samples | 0.29±.08 | **0.02**±.01 | **0.01**±.00 | **1.00**±.00 | 0.16±.05 | **0.01**±.01 | **1.00**±.00 | 0.73±.08 | **0.01**±.01 |

### E.2    REPLICATION OF WAIC RESULTS

We did not include WAIC because we were not able to replicate the results of Choi et al. (2019). The figure to the right shows a WAIC histogram for CIFAR-10 (blue) vs SVHN (OOD, orange) computed using our Glow implementation (ensemble size 5). We attempted to reproduce Choi et al.'s Figure 3, which shows SVHN having lower and more dispersed scores than CIFAR-10. We did not observe this: all SVHN WAIC scores overlap with or are higher than CIFAR-10's, meaning that SVHN can not be distinguished as the OOD set. Two differences between our Glow implementation and theirs were that they use Adam (vs RMSprop) and early stopping on a validation set. We found neither difference affected results.

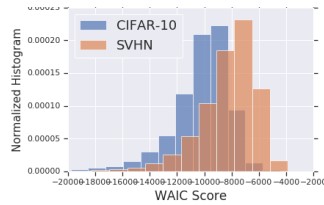

### E.3 VARYING $M$ FOR GLOW

Figure 4 reports results for our typicality test on Glow, varying $M$ from $[1, 150]$. Table 2's results are a subset of these. We also report evaluations using CIFAR-100 as an OOD set.

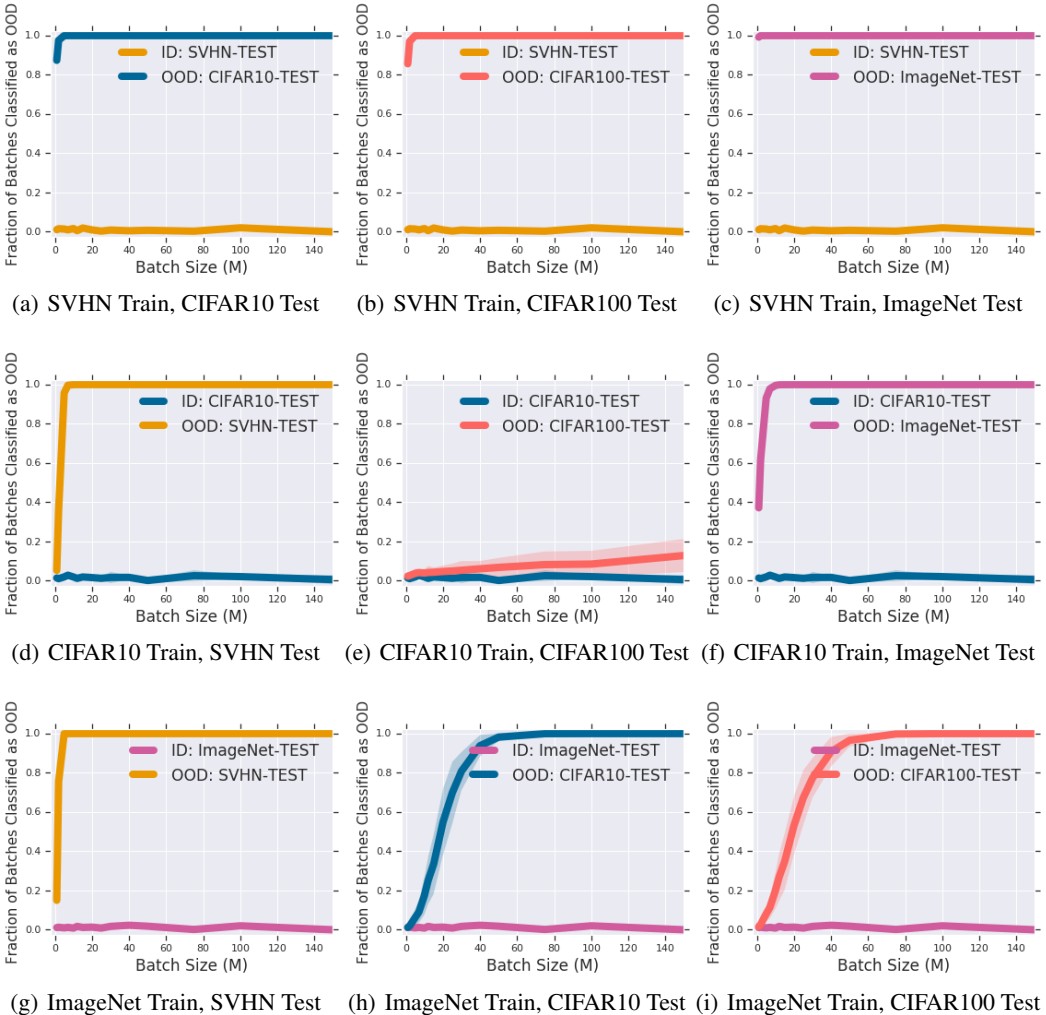

Figure 4: *Natural Image OOD Detection for Glow*. The above plots show the fraction of $M$-sized batches rejected for three Glow models trained on SVHN, CIFAR-10, and ImageNet. The OOD distribution data sets are these three training sets as well as CIFAR-100.

