# OpenReview forum: "Detecting Out-of-Distribution Inputs to Deep Generative Models Using Typicality"
_ICLR.cc/2020/Conference — Reject_

### Official Review · AnonReviewer2 · 2019-10-20
**Official Blind Review #2**

**Rating:** 3

**Review:**

I've read the authors' rebuttal and other reviews; I'd like to keep my score as is. My main concerns are the novelty of the work, the theoretical soundness of the method for small data settings and its robustness in settings with model misspecification.

#################################

The paper proposes a new approach based on the notion of typical set in probability and tackles the challenging problem of detecting OOD using deep generative models. The main claim of the paper is that assigning high likelihood to OOD samples in DGMs is due to the mismatch between model’s typical set and its high probability density areas.

I liked the idea of proposing a hypothesis testing approach for finding OOD samples generated from a model; however, my main concern is that the approach has some major practical limitation that the authors have also rightly mentioned in their discussion. It seems that even with a hypothesis testing tool for OOD detection, the model capacity and other properties of the model are more fundamental and critical for OOD detection in DGMs. In other words, how this tool can be useful in practice if the models are misspecified and how robust is the tool with respect to model properties. This major limitation has not been addressed in the experiments.

This paper, does a good job in finding the OOD data points if the likelihood histograms do not overlap using the typicality notion. However, this idea had already been proposed and explored in Choi. et al. 2019 (although for a flow-based model). This makes the technical novelty of the work less significant.

Overall, I think the paper needs some improvement in terms of discussing the robustness of the test with respect to model properties; otherwise, it is just another typicality set explanation of why DGMs may produce high likelihood values for OOD samples which has already been mentioned in previous work.

**Experience Assessment:**

I have read many papers in this area.

**Review Assessment: Checking Correctness Of Derivations And Theory:**

I assessed the sensibility of the derivations and theory.

**Review Assessment: Checking Correctness Of Experiments:**

I assessed the sensibility of the experiments.

**Review Assessment: Thoroughness In Paper Reading:**

I read the paper at least twice and used my best judgement in assessing the paper.

---

> ### Author Response · Authors · 2019-11-11
> **Response to R2**
>
> Thank you for your comments, R2.  We are glad to hear that you “liked the idea of proposing a hypothesis testing approach” and appreciate that the method “does a good job in finding the OOD data points.”  We hope to address your doubts below.
>
> 1. Novelty w.r.t. Annulus Method:  It is incorrect to state that Choi et al. [2019] have previously “explored” a typicality-based solution.  Their annulus method is what our test reduces to in the *special case* of the $(\epsilon, M=1)$-typical set for isotropic Gaussians (which we state on p 6).  Choi et al. make no mention of the general entropy-based definition of typicality (our Def 2.1).  Nor do they give any methodology for testing the $(\epsilon, M>1)$-typical set in Gaussians, let alone any other class of deep generative model (eg PixelCNN, VAEs).   Previous to our work, the question of how to test for typicality *in every class of deep generative model except Gaussian flows* was completely wide open and unaddressed.
>
> 2. Behavior Under Misspecified Models:  While model misspecification is certainly a concern when building generative models, the topic of our paper is not model building.  Rather, we focus on testing a given, pre-trained generative model.  Or to be precise, we are testing $q$, some unknown distribution we observe only through samples, vs $p_{\theta}$, the pre-trained model.  Model misspecification tests $p^{*}$, the distribution of the training data, vs $p_{\theta}$ (or $p_{\theta}$ vs $p_{\theta’}$).  Clearly these are much different settings, making an analysis of misspecification well out-of-scope for our paper.
>
> Again, thank you for taking the time to read our draft.  We look forward to further discussion of these points.

---

### Official Review · AnonReviewer1 · 2019-10-23
**Official Blind Review #1**

**Rating:** 6

**Review:**

Thanks for the authors for your detailed reviews.

My major concerns about the proposed method are whether "the typicality set" could be faithfully applied in the small data regime. The authors point me to the interesting Figure 4, which shows that it basically achieves converged performance when $m = 50$ or smaller numbers for some problems. I think this experiment is a strong support for the proposed method.

However, I don't agree that the M=1 Gaussian case acts as a strong support for the method. As I said, for some other wired distribution, it is difficult to interpret what the M=1 Typicality set becomes.

The authors also clarify the difference between different baselines.

Overall, I will increase my score to "Weak Accept".

##########################

Recent works have shown that out-of-distribution samples can have higher likelihoods than in-distribution samples for some generative models. To explain this phenomenon and to tackle the problem for OOD detection, this paper adopts "typical sets" for identifying in-distribution samples. Specifically, a "typical set" is a set of examples whose expected log likelihood approximate the model's entropy. For a Gaussian distribution, the paper finds that a single point typical set locates exactly in the \sqrt{d} radius, which is usually favored over the high-likelihood origin. Then the paper uses the "typical set" to perform OOD for a batch of examples. Empirically they demonstrate competitive performance over MNIST and natural image tasks.

Typical set seems natural for out-of-distribution detection. An important property is that, if one draws a large number of independent samples from the distribution, it is very likely that these samples belong to the typical set (basically Theorem 2.1).  However,  for small n, this property doesn't hold anymore, which leaves here a questionmark whether "Typical set" can be used for OOD detection in small n regime. As the author argues, for Gaussian distribution when n=1 the typical locations are those \sqrt{d} radius points. But this doesn't justify the "Typical set". If the distribution is some non-Gaussian wired distribution, the typical locations doesn't seem to make sense at all.

Following the previous argument above, the Typical set method requires to perform OOD for a batch of examples. In contrast, the Annulus method can be directly applied to one single test example.

Empirically, the Typically set doesn't demonstrate obvious advantages compared to the baselines. For both MNIST and natural image tasks, it seems that all methods behave similarly. For comparing such big tables, I would recommend adding a column showing the average ranks among all methods. Beyond that, standard OOD tasks usually evaluate methods using AUROC and AUPR (Hendrycks and Gimpel, 2017). Is it possible to also include such metrics ?

Theorem 2.1 is confusing. It is beneficial to define what P is, and verbally state what the theorem conveys.

**Experience Assessment:**

I have read many papers in this area.

**Review Assessment: Checking Correctness Of Derivations And Theory:**

I carefully checked the derivations and theory.

**Review Assessment: Checking Correctness Of Experiments:**

I carefully checked the experiments.

**Review Assessment: Thoroughness In Paper Reading:**

I read the paper thoroughly.

---

> ### Author Response · Authors · 2019-11-11
> **Response to R1**
>
> Thank you for taking the time to read our paper, R1.  We are glad that you found our approach “natural for out-of-distribution detection.”  Please consider the following rebuttals to your critiques.
>
> 1. Small $M$ Behavior:  You are correct in that Thm 2.1 only holds for sufficiently large $M$, and therefore it is not obvious that a typicality-based test behaves correctly in the small-$M$ regime.  Firstly, we emphasize that GoF testing for deep generative models is still a completely open problem even in the large-M regime.  As KSD is the primary competitor and scales as $\mathcal{O}(M^{2})$, even a procedure that performs well for large M is a contribution.  And as can be seen in Figure 4, our test works near perfectly for $M>50$, and this prompted us to focus the main experimental section on the $M \in \{2, 10, 25 \}$ regime.
> Now returning to the main point, we have two pieces of evidence for the test’s correctness for small $M$.  The first is theoretical: examining the Gaussian case, the $(\epsilon=0, M=1)$-typical set is $\mathcal{A}_{0}^{1}[N(x;\mu, \sigma)] = \{x \mid || x - \mu ||_{2} = \sigma \sqrt{d} \}$, the shell of radius $\sigma \sqrt{d}$.  This is quite a narrow region of the support, which speaks to the inapplicability of Thm 2.1.  Yet as we are given only one test instance, such a restriction seems to be the most appropriate choice.  It is clearly better than picking the mode, for instance.  Our second piece of evidence is experimental: Our test performs reasonably well in the $M=10$ regime and in cases can achieve near perfect OOD classification for $M=2$.  In the case of Glow trained on SVHN, 98% of CIFAR-10 batches and 100% of ImageNet batches were correctly classified at $M=2$.  Hence, our test must be checking sensible regions of data space or such good results could not be possible.  In turn, we find the claim that “the typical locations doesn't seem to make sense at all [for small $M$]” completely speculative.  Can you please provide some reasoning in support of your claim?
>
> 2.  Performance in Comparison to Baselines:  Firstly, let us clarify that the only relevant baseline from the goodness-of-fit-testing literature is the Kernelized Stein Discrepancy (KSD).  As stated in Sec 3.1, we are aware of no other GoF test that can be widely applied across all types of deep generative models.  As for our test vs KSD, they perform roughly the same in all cases except for the PixelCNN trained on FashionMNIST.  KSD is unable to detect MNIST as OOD, and our test is unable to detect NotMNIST as OOD.  Yet an additional factor that differentiates the two is runtime: KSD is drastically slower, requiring an $\mathcal{O}(dM^{2})$ evaluation time (with M representing the batch size) in addition to the cost of computing derivatives through the model.  Our method is $\mathcal{O}(M)$ after the likelihoods have been computed from the model.  With runtime considered, the results clearly favor our method.
> Secondly, the t-Test and KS-test baselines are not really ‘competitors’ as they are (i) proposed by us and (ii) closely based on our typicality test.  The t-test is just the typicality test with $\epsilon = 0$, and the KS-test is comparing all moments, whereas our typicality test is comparing just the first.  Thus, these tests should perform comparably!  The fact that the KS-test and ours perform so similarly can be seen as positive evidence for our method since it validates that the first moment (i.e. entropy) is truly the critical one for testing OOD.
> Thirdly, the Maximum Mean Discrepancy (MMD) baseline is not a `valid’ competitor as it is performing a two-sample test, not a GoF test, which is the topic of our paper.  We provide MMD only as a reference point to see how testing against $p^{*}$ compares to testing against $p_{\theta}$.  Lastly, the annulus method [Choi et al., 2019] is simply what our test reduces to in the special case of the $(\epsilon, M=1)$-typical set for isotropic Gaussians (which we state on p 6).
>
> 3.  “The Typical set method requires to perform OOD for a batch of examples. In contrast, the Annulus method can be directly applied to one single test example.”:  Pitting the annulus method vs our typicality test is a false dichotomy.  The annulus method (as mentioned above) is a special case of our test, and the only reason it can be used for one-sample is that it’s assuming the $(\epsilon, M=1)$-typical set.
>
> 4.  AUROC metrics:  As far as we are aware, AUROC metrics in the literature are computed for point-wise rejection rules.  Thus they would not be comparable to our hypothesis testing / batch-wise methodology, which we believe to be the first of its kind for deep generative models.  Yet, we thank the reviewer for the suggestion, and we will look into adding ROC-based metrics to our revised draft.
>
> 5.  Thm 2.1:  Thank you for the feedback.  We will add more discussion of the intuition.
>
> Again, thanks for sharing your thoughts.  We look forward to further discussion.

---

### Official Review · AnonReviewer3 · 2019-10-24
**Official Blind Review #3**

**Rating:** 6

**Review:**

This paper is concerned with how to determine whether a set of data points are from a given distribution. It uses the so-called typical set to transform the problem to determining whether the data points lie in the typical set of the given distribution. It proposes a statistical test using the empirical distribution of model likelihoods to determine whether inputs lie in the typical set if the considered model.

The motivation of the work is very clear, and the paper is well organized. The basic idea of using the typical set to check whether given data points are from a given distribution seems sensible, as guaranteed by Theorem 2.1.

My concern is about the performance of the proposed method compared to alternatives. First, a standard approach to the considered problems seems to be the two-sample tests (or its approximations or variations), so it would be desirable to compare the typical set-based approach with the two-sample test approaches theoretically. In particular, given that you have to allow some error when using typical sets, what is exactly the advantage of the proposed approach?  Second, according to the empirical results (Section 5), the proposed method does not seem to clearly outperform alternatives such as KS-test. In this case, a better justification of the reliability of the proposed approach would be helpful.

I acknowledge I read the authors' response and other reviews and would like to keep my original rating. (I agree that the t-Test and KS-test were probably first used by the authors, but at the same time it is natural to adopt them; that is why I considered them as baselines.)

**Experience Assessment:**

I have read many papers in this area.

**Review Assessment: Checking Correctness Of Derivations And Theory:**

I carefully checked the derivations and theory.

**Review Assessment: Checking Correctness Of Experiments:**

I assessed the sensibility of the experiments.

**Review Assessment: Thoroughness In Paper Reading:**

I read the paper thoroughly.

---

> ### Author Response · Authors · 2019-11-11
> **Response to R3**
>
> Thank you for your feedback, R3.  We are glad that you found the work’s motivation “very clear” and the paper “well organized.”  Hopefully we can dispel your reservations below.
>
> 1. Performance in Comparison to Baselines:  Firstly, let us clarify that the only relevant baseline from the goodness-of-fit-testing literature is the Kernelized Stein Discrepancy (KSD).  As stated in Sec 3.1, we are aware of no other GoF test that can be widely applied across all types of deep generative models.  As for our test vs KSD, they perform roughly the same in all cases except for the PixelCNN trained on FashionMNIST.  KSD is unable to detect MNIST as OOD, and our test is unable to detect NotMNIST as OOD.  Yet an additional factor that differentiates the two is runtime: KSD is drastically slower, requiring an $\mathcal{O}(dM^{2})$ evaluation time (with M representing the batch size) in addition to the cost of computing derivatives through the model.  Our method is $\mathcal{O}(M)$ after the likelihoods have been computed from the model.  With runtime considered, the results clearly favor our method.
> Secondly, the t-Test and KS-test baselines are not really ‘competitors’ as they are (i) proposed by us and (ii) closely based on our typicality test.  The t-test is just the typicality test with $\epsilon = 0$, and the KS-test is comparing all moments, whereas our typicality test is comparing just the first.  Thus, these tests should perform comparably!  The fact that the KS-test and ours perform so similarly can be seen as positive evidence for our method since it validates that the first moment (i.e. entropy) is truly the critical one for testing OOD.
> Thirdly, the Maximum Mean Discrepancy (MMD) baseline is not a `'valid' competitor as it is performing a two-sample test, not a GoF test.  We provide MMD only as a reference point to see how testing against $p^{*}$ compares to testing against $p_{\theta}$.  Lastly, the annulus method [Choi et al., 2019] is simply what our test reduces to in the special case of the $(\epsilon, M=1)$-typical set for isotropic Gaussians (which we state on p 6).
>
> 2.  “A standard approach to the considered problems seems to be the two-sample tests”: This is incorrect.  Two-sample tests assume the setting $q$ vs $p^{*}$, with $p^{*}$ being the inaccessible underlying generative process.  Rather, we are testing  $q$ vs $p_{\theta}$, with $p_{\theta}$ being a model to which we have access.  Yet we do report an MMD baseline in the experiments in anticipation of readers wondering how a two-sample test would perform in the same setting.  We find it to perform comparably except in the same setting mentioned for KSD above: for the PixelCNN trained on FashionMNIST, MMD is unable to detect MNIST as OOD, and our test is unable to detect NotMNIST as OOD.  Note that this performance was only able to be achieved when we derived the MMD kernel from the generative model.  Otherwise, performance was strictly worse than our test.   Moreover, MMD is much more expensive to compute as it is $\mathcal{O}(MNd)$ (as mentioned on p 6).
>
> 3.  “Given that you have to allow some error when using typical sets, what is exactly the advantage of the proposed approach?”:  We are not sure what you mean exactly by “allow for some error.”  But to summarize our contributions once more: (1) we propose typicality as an explanation for the OOD phenomenon in deep generative models, (2) derive a GoF test widely applicable across all deep generative model classes and with better runtime than KSD, (3) show that this test, despite it using only the first moment of the empirical likelihoods (which makes runtime cheap), is able to detect the OOD set in many of the cases reported by [Nalisnick et al., 2019].
>
> Again, thanks for taking the time to read our paper.  We look forward to further discussion.

---

### Public Comment · ~Shengyu_Zhu1 · 2019-10-18
**Interesting work. Some questions about the motivation example and the problem of testing typicality.**

Dear authors,

It is interesting to see another paper on typicality on ICLR. Yet I don't quite understand the motivation example for typicality, and also the problem of testing typicality.

Typical set indicates that a small set, compared to the whole support, takes most of the probability (e.g., when considering finite sample space, this can be understood as number of different sample sequences). In my experience, typicality is meaningful if you have a large number of i.i.d. samples so that law of large numbers can be used (Theorem 2.1 in your manuscript) and $\epsilon$ defining the typical set is small so that sample sequences contained in the $\epsilon$-typical set is 'close' to be typical. In this sense, it is quite strange to consider only one or few samples when talking about typicality. For the Gaussian example with $\mathcal N(0, I)$ with high dimension, as you also point out in footnote, it is OK to consider so because each dimension can be considered independent and has the same marginal distribution. For images, this does not hold true: different pixels are not independent and are very likely to have different marginal distributions.

For the problem defined in Eq.~(2): if the deviation parameter $\epsilon$ is large, then the sample sequence in a typical set is not that 'typical'; on the other hand, if $\epsilon$ is small and the number of sample is also small, say,  $M=25$  (which I found to be the largest number considered in this paper; correct me if I'm wrong), AEP or law of large numbers cannot be used and the probability of typical set can be low, which seems somewhat contradictory to the meaning of typicality.

By the way, in Section 3.2., the condition that $\mathcal A_{\epsilon}[q(x)]$ is not a subset of $\mathcal A_{\epsilon}[p(x)]$, with your proof in Appendix A.2 where $M$ is sufficiently large,  seems equivalent to say that $p$ and $q$ do not have the same entropy: with $\epsilon <  |H(p)-H(q)|/2$, this adopted condition in the paper hold with high probability when $M \to\infty$.

---

> ### Author Response · Authors · 2019-10-21
> **Author Response**
>
> Hi Shengyu,
>
> Thanks for taking the time to read our paper and share your thoughts.  It’s much appreciated.
>
> I’m not sure that we follow your line of reasoning about the small-$M$ regime.  Let’s return to the $M=1$ Gaussian example but now with d=1 (with $d$ denoting dimensionality) so that there’s no dimension-wise factorization to consider.  In this case the test would simplify to: $ \mid \sigma^{2} - (x - \mu)^{2} \mid \le 2 \sigma^{2} \epsilon$.  Intuitively this expression is testing if $x$’s distance from the mean is roughly $\sigma$, with the bound relaxing as a function of $\sigma$.  This seems like reasonable behavior to us.  One interesting stress test to consider is the case of $x \approx \mu$, which could easily happen in the $d=1$ case.  We’d then have to set $\epsilon \ge \frac{1}{2}$ for the bound to hold.  Yet still $\epsilon = 1/2$ does not seem like an unreasonably high setting for $\epsilon$ that would signal that our test is problematic.
>
> In general, we don’t agree that “it is quite strange to consider only one or few samples when talking about typicality.”  You seem to imply that $\epsilon$ must be set large, but this is not the case.  Rather, it’s just that the criterion for what is a ‘typical’ batch of samples becomes more stringent.  To see this, consider the general bound $ \mid \frac{1}{M} \sum_{m=1}^{M} -\log p(x_{m}) - \mathbb{H}[p]  \mid \le \epsilon$ (Equation 3).  For small $M$, this means that those few samples must approximate the entropy very well for the test to pass.  In the Gaussian example (for general $d$), it must be that all $x_{m}$’s fall very close to the annulus radius.  For large $M$, a few samples can be far from the annulus just so long as the mean log density is still close to the entropy---hence the small-$M$ case being ‘more stringent’ as there is more contribution from any given sample.
>
> As for your comment on the proof, we appreciate you taking the time to read even the appendix.  However, we don’t quite agree that the subset condition is equivalent to saying the distributions $p$ and $q$ have the same entropy.  Consider the case $p=N(\mu=-1000, \sigma=1)$ vs $q=N(\mu=1000, \sigma=1)$.  These distributions have the same entropy, but clearly $\mid \mathbb{E}_{q}[ -\log p(x)] - \mathbb{H}[p]  \mid >> 0$.
>
> Lastly, we do indeed have results for $M > 25$.  Please see Appendix E.3, Figure 4 for results showing when $M$ is as large as $150$.  Our procedure achieves near perfect discrimination of the OOD set on almost all of the dataset pairs (which, of course, is due to the AEP kicking in, as you mention). Since our method achieved excellent OOD performance for high-values of $M$, we thought the more interesting and practically relevant experimental question was to understand how OOD detection changes as $M$ is decreased.  That’s why we focused on $M \le 25$ in the main experimental section.  We will update the text to clarify this.

---

> > ### Public Comment · ~Shengyu_Zhu1 · 2019-10-22
> > **Thanks for your response**
> >
> > Thanks for your quick response. It was my bad to misunderstand the proof: it is $\mathbf E_q[-\log p(x)]$ which I mistakenly thought as $\mathbf E_q[-\log q(x)]$. And also I did not look at the appendix for more experiments. So thanks again for your response.
> >
> > I agree with you that a small $\epsilon$ can give more stringent condition, but still the AEP cannot be used with a small number $M$. Of course AEP only gives a lower bound. Here you want to show that even with small number the probability of typical set (through In-Distribution) is also high. Is this correct?

---

> > > ### Author Response · Authors · 2019-10-22
> > > **Requesting Clarification**
> > >
> > > Thanks for continuing the discussion, Shengyu.  We agree that the AEP cannot be used when $M$ is small.  However, we are not sure of exactly what you're asking in your last statement.  Are you referring to Thm 2.1: $P(\mathcal{A}^{N}_{\epsilon}[p(x)]) > 1 - \epsilon$ (with $N$ now serving for $M$)?

---

> > > > ### Public Comment · ~Shengyu_Zhu1 · 2019-10-22
> > > > **More about question**
> > > >
> > > > Once again, thanks for your prompt response. It was regarding your last paragraph. Let me quote this sentence right after Eq. (2):
> > > >
> > > > 'The intuition is that if $\tilde X$ indeed sampled from $p_\theta$, then with high probability it must reside in the typical set (Theorem 2.1)'.
> > > >
> > > > AEP or Theorem 2.1 requires $N$ or here $M$ to be large enough, or let's say 'sufficiently large'. Then the probability of the typical set is $> 1-\epsilon$ for any fixed $\epsilon>0$. The exact number being 'sufficiently large' depends on the distribution and also $\epsilon$. My question is, with $M=150$ or any given number, is this $M$ indeed 'sufficiently large' so that the probability of the typical set is high? I don't think that this statement can be verified by the AEP or Theorem 2.1, but the experiment with In-Distribution testing should be able to.
> > > >
> > > > Please let me know if question is clear now and also if my understanding is correct. Thanks.

---

> > > > > ### Author Response · Authors · 2019-10-22
> > > > > **Thanks for the Clarification**
> > > > >
> > > > > Thanks for the clarification.  I think we understand your point now (but please correct us if not).
> > > > >
> > > > > As for at what value of $M$ does the AEP kick-in, it must do so at $M > 50$ since we observe that type-I and type-II error are essentially nonexistent in nearly all cases tested.  For $M \le 50$, it’s hard to say.  Please let us know if you have any ideas about how to directly quantify the transition.
> > > > >
> > > > > Yet you do make a good point about Thm 2.1 in the small $M$, small $\epsilon$ regime.  Theorem 2.1 indeed does not apply in this regime, and the sentence that you quoted has the potential of being interpreted too generally.  Yet we believe it’s important to emphasize (for other readers, at least) that just because Thm 2.1 doesn’t apply here, that doesn’t mean that the procedure is ill-behaved.  Returning to the isotropic Normal example once more, the $(\epsilon=0, M=1)$-typical set is $\mathcal{A}_{0}^{1}[N(x;\mu, \sigma)] = \{x \mid || x - \mu ||_{2} = \sigma \sqrt{d} \}$, the shell of radius $\sigma \sqrt{d}$.  Clearly this set is a poor summary of what we expect to be generated by $N(x;\mu, \sigma)$, which speaks to the inapplicability of Thm 2.1.  But since we are testing only one sample ($M=1$), it makes sense that our procedure would become conservative, only letting the test pass if $\tilde{x}$ falls in a very restricted region.  We will incorporate this discussion into the paper during the next revision.  Thanks again for the discussion.

---

> > > > > > ### Public Comment · ~Shengyu_Zhu1 · 2019-10-22
> > > > > > **Thanks again**
> > > > > >
> > > > > > Now I think I understand the motivation example, which is indeed insightful and interesting.
> > > > > >
> > > > > > I don't think it is easy to quantify the transition, either. One way is to get closed form of the probability of the typical set for any finite $M$, when $p$ is given. But of course, this can be pretty hard. Is it simpler to get a somewhat tight lower bound? Or using experiments (like Monte-Carlo) for given $p$ to estimate it? Just random thoughts, but this is a good question to think about. Actually in many information theory problems, one really does not care about the exact number to be 'sufficiently large'. For example, in lossless source coding, Shannon allowed the code length to be infinite.
> > > > > >
> > > > > > Nice work!

---

### Decision · Program_Chairs · 2019-12-19

**Decision:**

Reject

**Comment:**

This paper tackles the problem of detecting out of distribution (OoD) samples. To this end, the authors propose a new approach based on typical sets, i.e. sets of samples whose expected log likelihood approximate the model's entropy. The idea is then to rely on statistical testing using the empirical distribution of model likelihoods in order to determine whether samples lie in the typical set of the considered model. Experiments are provided where the proposed approach show competitive performance on MNIST and natural image tasks.

This work has major drawbacks: novelty, theoretical soundness, and robustness in settings with model misspecification. Using the typicality notion has already been explored in Choi. et al. 2019 (for flow-based model), which dampers the novelty of this work. The conditions under which the typicality notion can be used are also not clear, e.g. in the small data regime. Finally, the current experiments are lacking a characterization of robustness to model misspecification. Given these limitations, I recommend to reject this paper.